# FARTrack: Fast Autoregressive Visual Tracking with High Performance

**Guijie Wang**[1,2*], **Tong Lin**[1,2*], **Yifan Bai**[3*‡], **Anjia Cao**[1,2],
**Shiyi Liang**[1,2], **Wangbo Zhao**[4], **Xing Wei**[1,2†]

[1]School of Software Engineering, Xi'an Jiaotong University
[2]State Key Laboratory of Human-Machine Hybrid Augmented Intelligence, Xi'an Jiaotong University
[3]DAMO Academy, Alibaba Group     [4]National University of Singapore
{guijiewang,lintong1220,caoanjia7,sy_liang2023}@stu.xjtu.edu.cn
baiyifan.byf@alibaba-inc.com   wangbo.zhao96@gmail.com
weixing@mail.xjtu.edu.cn

## Abstract

Inference speed and tracking performance are two critical evaluation metrics in the field of visual tracking. However, high-performance trackers often suffer from slow processing speeds, making them impractical for deployment on resource-constrained devices. To alleviate this issue, we propose **FARTrack**, a **F**ast **A**uto-**R**egressive **T**racking framework. Since autoregression emphasizes the temporal nature of the trajectory sequence, it can maintain high performance while achieving efficient execution across various devices. FARTrack introduces **Task-Specific Self-Distillation** and **Inter-frame Autoregressive Sparsification**, designed from the perspectives of **shallow-yet-accurate distillation** and **redundant-to-essential token optimization**, respectively. Task-Specific Self-Distillation achieves model compression by distilling task-specific tokens layer by layer, enhancing the model's inference speed while avoiding suboptimal manual teacher-student layer pairs assignments. Meanwhile, Inter-frame Autoregressive Sparsification sequentially condenses multiple templates, avoiding additional runtime overhead while learning a temporally-global optimal sparsification strategy. FARTrack demonstrates outstanding speed and competitive performance. It delivers an AO of 70.6% on GOT-10k in real-time. Beyond, our fastest model achieves a speed of 343 FPS on the GPU and 121 FPS on the CPU. Source code is available at: https://github.com/MIV-XJTU/FARTrack.git

## 1 Introduction

Visual object tracking (VOT), aiming to continuously localize arbitrary objects in a video sequence, relies on the continuous positions of the objects and is highly sensitive to temporal information Asanomi et al. (2023); Mayer et al. (2022); Wu et al. (2023b); Zhao et al. (2023); Zhou et al. (2023a). In practical applications on edge devices with limited resources, it is often necessary to consider both speed and performance simultaneously Yang et al. (2026); Zeng et al. (2025b;a); Qi et al. (2026). However, existing methods can only achieve either high speed Gopal & Amer (2024); Li et al. (2023); Xie et al. (2023); Yang et al. (2023b); Zaveri et al. (2025) or high performance Cai et al. (2024); Chen et al. (2023); Hong et al. (2024); Xie et al. (2024); Yang et al. (2023a).

To address this dilemma, existing efforts to balance the tracking speed and performance can be broadly categorized into two approaches:

*(i)* Model distillation methods Cui et al. (2023); Guo et al. (2021); Hinton et al. (2015); Li et al. (2017); Romero et al. (2014) based on cross-layer train a student model to mimic a teacher's vision features. However, as shown in Figure 1(a), these methods rely on **hand-crafted layer assignments** to enable knowledge transfer Ahn et al. (2019); Passalis et al. (2020); Romero et al. (2014); Tung & Mori (2019); Yue et al. (2020); Zagoruyko & Komodakis (2016). Without prior knowledge of

---

*Equal contribution. †Corresponding author. ‡Project Lead

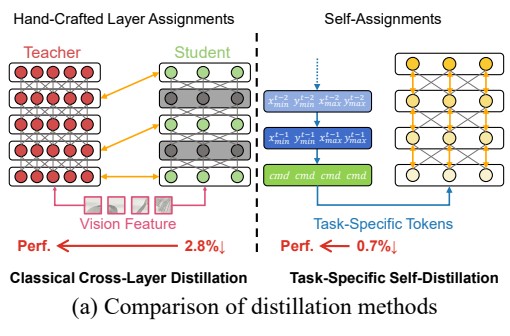

(a) Comparison of distillation methods

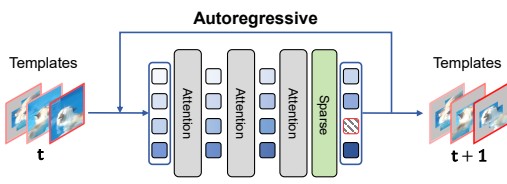

(b) Inter-frame Autoregressive Sparsification

Figure 1: **Overview.** (a) Comparison of our Task-Specific Self-Distillation and Classical Cross-Layer Distillation. (b) Inter-frame Autoregressive Sparsification for Multi-templates.

Figure 2: **FARTrack vs. Other Trackers: Performance-Speed Trade-off.** Comparison of our FARTrack with the state-of-the-art trackers on GOT-10k in terms of tracking speed (horizontal axis) on GPU and AO performance (vertical axis). The diameter of the circle is proportional to the ratio of the model's speed to its performance. FARTrack$_{naco}$ significantly surpasses existing trackers in both tracking performance and inference speed.

teacher-student layer pair assignment, manually designed ones often disrupt the hierarchical structure of feature extraction, thereby failing to achieve optimal results. Moreover, the distillation objectives of these methods focus on current-frame visual features, overlooking temporal information in trajectory sequences critical for tracking tasks.

*(ii)* Runtime token sparsification approaches Chen et al. (2022b); Liang et al. (2022); Rao et al. (2021); Ye et al. (2022) typically involve the gradual removal of a subset of tokens during inference. However, this process introduces **extra computational overhead** for identifying tokens to remove, ultimately reducing tracking efficiency. Moreover, as these methods prioritize the current frame rather than the entire frame sequence, they fail to achieve a temporally-global optimal solution, which adversely impacts overall tracking performance.

To address these two issues, we present a fast, high-performance multi-template autoregressive framework for visual tracking, using multi-template design to boost accuracy. Our framework comprises two key components: *(i)* **Task-Specific Self-Distillation**. Unlike classical cross-layer distillation, our approach conducts layer-by-layer distillation of task-specific tokens, which represent the object's trajectory sequences. In this method, each layer acts as both student and teacher for the next, trained to fit teacher's trajectory sequence features via KL divergence. Our approach avoids suboptimal manual layer assignments while maintaining temporal information. *(ii)* **Inter-frame Autoregressive Sparsification**. Compared with the frame-wise runtime sparsification methods, our sparsification is a sequence-level method for template sequences. We treat attention weights as matrices to retain foreground tokens while discarding background tokens, and propagate sparsification results to subsequent frames in an autoregressive manner. In this process, we reduce the bandwidth without introducing extra computational load, while retaining temporal information to learn a temporally-global optimal sparsification strategy.

Overall, we present **FARTrack**, a fast and high-performance tracking framework. Extensive experiments demonstrate the effectiveness and efficiency of our approach. Our method achieves a better balance between inference speed and tracking performance than previous trackers. Specifically, as demonstrated in Figure 2, compared to the high-performance tracker AsymTrack-B, FARTrack$_{tiny}$ attains a 2.9% higher AO score on the GOT-10k benchmark while achieving comparable running speed on the GPU. Moreover, FARTrack$_{pico}$ delivers 0.5% better performance than AsymTrack-T on GOT-10k, along with superior GPU (343 FPS) and CPU (121 FPS) speeds.

## 2 RELATED WORK

**Efficient Tracking Framework.** In practical application scenarios, it is imperative to deploy trackers that achieve both high speed Cai et al. (2023); Kou et al. (2023); Li et al. (2023); Wei et al. (2024); Zhang et al. (2023); Zhou et al. (2023b); Zhang et al. (2026) and high performance Gao et al. (2023); Li et al. (2023); Shi et al. (2024); Tang et al. (2024); Wu et al. (2023a); Zheng et al. (2024) on resource-constrained edge devices. Over the past decade, researchers have been exploring efficient and effective tracking framework for real-world applications Bhat et al. (2019); Cao et al. (2022); Danelljan et al. (2019; 2017); Sun et al. (2025); Xie et al. (2022); Xu et al. (2020). While Siamese trackers Bertinetto et al. (2016); He et al. (2023); Li et al. (2019; 2018); Shen et al. (2022); Tang & Ling (2022); Xing et al. (2022); Zaveri et al. (2025) with lightweight designs Yan et al. (2021b) or dynamic updates Borsuk et al. (2022) reduce computation, they often overlook temporal dependencies, limiting performance. Transformer-based methods Blatter et al. (2023); Chen et al. (2023); Gao et al. (2022; 2023); Kang et al. (2023); Lin et al. (2022); Song et al. (2023; 2022); Ye et al. (2022); Zhang et al. (2022) improve accuracy but add complexity through decoding heads. Recent generative paradigms Bai et al. (2024); Chen et al. (2023); Wei et al. (2023a) eliminate custom heads yet incur high computational costs. Existing frameworks thus face trade-offs between efficiency and performance. In this paper, we propose a more efficient generative tracking framework to better balance speed and performance.

**Model Distillation.** Model distillation Ahn et al. (2019); Cui et al. (2023); Shen et al. (2021); Tung & Mori (2019); Wu et al. (2024); Ma et al. (2025); Cao et al. (2025); Du et al. (2025) transfers knowledge from a teacher to a lightweight student. Typical methods like AVTrack Wu et al. (2024) and MixformerV2 Cui et al. (2023) use multi-teacher maximization or layer skipping. However, such cross-layer distillation Wang et al. (2024); Zhang et al. (2023) often relies on suboptimal manual layer associations, leading to notable performance drops. Our approach compresses the model via self-distillation on task-specific tokens between adjacent layers, avoiding manual pair assignments and preserving temporal information.

**Token Sparsification.** Existing methods Chen et al. (2022b); Liang et al. (2022); Rao et al. (2021); Ye et al. (2022); Zhao et al. (2024b;a; 2025a;b; 2026) reduce computation by progressively removing less important tokens during runtime. DynamicViT Rao et al. (2021) employs lightweight predictors for stepwise token pruning, while OSTrack Ye et al. (2022) removes background regions early in processing. However, such runtime approaches often introduce extra steps, increasing latency, and focus only on the current frame. We propose a sequence-level post-processing sparsification method that avoids additional runtime overhead, improves speed, and maintains high performance.

## 3 METHOD

### 3.1 REVISITING ARTRACK

ARTrack Wei et al. (2023a) is an end-to-end sequence generation framework for visual tracking, which represents object trajectories as discrete token sequences using a shared vocabulary. By quantizing discrete token items, we obtain the coordinates corresponding to each token, thereby enabling the model to depict object positions via discrete tokens. The framework then employs a Transformer Encoder to extract visual information and progressively model the sequential evolution of the trajectory prompted by the preceding coordinate tokens. ARTrackV2 Bai et al. (2024) adds a dynamic appearance reconstruction process on the basis of its predecessor. It models the trajectory while reconstructing the appearance in an autoregressive manner.

**Motivation.** Although the ARTrack series models maintain temporal information retention, their architectures incorporating excessive depth and numerous parameters exhibit bandwidth-unfriendly characteristics, ultimately reducing tracking efficiency. Conventional optimization methods, such as cross-layer distillation and runtime token sparsification, have been employed to mitigate these structural bottlenecks. However, cross-layer distillation relying on hand-crafted layer assignments disrupts the hierarchical structure of feature extraction and runtime token sparsification introduces extra computational overhead while neglecting temporal-global optimization within frame sequences. To address these limitations, we propose FARTrack, which reduces model depth via task-specific self-distillation for model compression and introduces an inter-frame autoregressive sparsification method to eliminate background redundancy and noise in the template images.

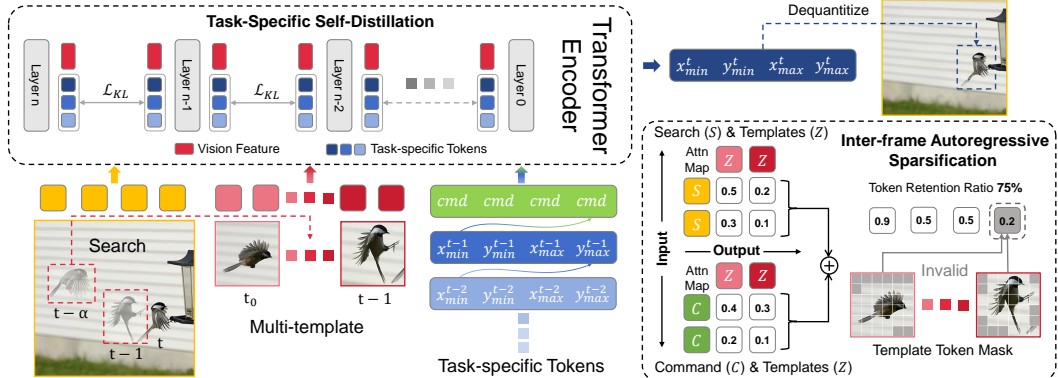

Figure 3: **FARTrack Framework.** FARTrack is a fast, high-performance multi-template autoregressive framework, comprising two key components: Task-Specific Self-Distillation for model compression and Inter-frame Autoregressive Sparsification for template sequences.

## 3.2 NETWORK ARCHITECTURE

As presented in Figure 3, the Transformer Encoder Dosovitskiy et al. (2020); He et al. (2022) encodes features and predicts the four coordinate tokens of the bounding box in an inter-frame autoregressive manner. Initially, all templates and the search image are divided into patches, flattened, and projected into a sequence of token embeddings. Subsequently, FARTrack maps the object positions across frames to a unified coordinate system with a shared vocabulary Chen et al. (2021; 2022a); Wei et al. (2023b), forming object trajectory tokens. Then, we concatenate the visual tokens, trajectory tokens, and four command tokens (representing the target bounding box coordinates), and input them into the Transformer Encoder. Finally, the Transformer encoder models the trajectory evolution in an autoregressive manner by leveraging the preceding trajectory tokens.

**Multi-templates.** To enhance tracking accuracy, we employ a multi-template design, further supported by a linear update strategy. To prevent the potential loss of temporal information caused by occlusion or disappearance of the target during the tracking process, we ensure that the updated multi-templates always include the first frame and the preceding frame.

## 3.3 TASK-SPECIFIC SELF-DISTILLATION

Knowledge distillation-based model compression reduces the model size, thereby improving tracking efficiency. However, current cross-layer distillation methods rely on suboptimal manual layer assignments Ahn et al. (2019); Cui et al. (2023); Shen et al. (2021); Tung & Mori (2019); Wu et al. (2024), disrupting hierarchical feature extraction and causing accuracy degradation in shallow models, while also neglecting temporal information in trajectory sequences.

To address this issue, we propose a simple yet effective model compression method known as task-specific self-distillation, as depicted in Figure 3. In our method, one model layer serves as the student layer and the corresponding next layer acts as the teacher layer, establishing layer-wise self-distillation Hou et al. (2024); Zhang et al. (2021; 2019) that inherently circumvents suboptimal manual layer assignments. Furthermore, our method operates on task-specific tokens which represent the object's trajectory sequences. The student layer is trained to fit the trajectory sequence features of the teacher layer by minimizing the KL divergence. Therefore, the temporal information in the trajectory sequences propagates backward among the layers, enabling the model to be distilled to a shallow level while maintaining accuracy. Ultimately, our method improves the tracking speed of the model while maintaining the tracking performance.

## 3.4 INTER-FRAME AUTOREGRESSIVE SPARSIFICATION

The template image contains target object features alongside persistent background and noise interference that reduces tracking efficiency. Traditional sparsification methods Chen et al. (2022b); Liang et al. (2022); Rao et al. (2021); Ye et al. (2022) prioritize runtime token elimination, but

these approaches introduce redundant time overhead and frame-specific optimization rather than sequence-level processing, resulting in slower speed and lower performance.

To eliminate template redundancy and noise in the tracking process while achieving a temporally-global optimal sparsification strategy, we propose an inter-frame autoregressive sparsification, as visualized in Figure 3. After processing through the attention layers, we obtain the attention weights for template tokens with respect to search tokens and the four command tokens based on the attention map. This sequential processing method considers the template's correlation with both the search area and the predicted coordinates, respectively. We then simply add these two attention weights and retain the parts with the largest values based on the predefined token retention rate. This process leverages the inter-frame correlations to mask the unnecessary background parts in the templates while retaining the key foreground parts. Subsequently, the sparsification results of the current frame are saved and propagated to subsequent frames in an autoregressive manner, thereby learning a temporally-global optimal sparsification strategy. Overall, our method requires no additional time-consuming processes and meanwhile preserves the temporal information, achieving faster speed and better performance.

Notably, masked tokens are excluded from processing to avoid distortion of normalization statistics. LayerNorm is exclusively applied to valid tokens, preventing improper scaling and shifting caused by statistical deviations that could undermine both inference stability and model performance.

## 3.5 Training and Inference

FARTrack is a fast tracker that enhances tracking efficiency by reducing model depth via task-specific self-distillation and eliminating redundant and noisy information from the templates using an inter-frame autoregressive sparsification.

**Training.** Similar to its predecessor, FARTrack undergoes both frame-level and sequence-level training Bai et al. (2024); Kim et al. (2022); Wei et al. (2023a); Liang et al. (2025). Initially, task-specific self-distillation is employed to progressively transfer trajectory sequence features from deeper layers to shallower layers, thereby reducing the model depth and achieving model compression. To ensure effective knowledge transfer, we minimize the KL divergence between the teacher and student layers during training. Building upon this model compression, we introduce an inter-frame autoregressive sparsification that computes a mask matrix based on inter-frame correlations and removes unimportant template background tokens according to a predefined token retention ratio.

Moreover, we incorporate the SIoU loss Gevorgyan (2022) to better capture the spatial correlation between the predicted and ground truth bounding boxes. For each video clip, the initial trajectory prompt is initialized using the object bounding box from the first frame and is propagated into subsequent frames in an autoregressive manner. The overall loss function is formulated as:

$$\mathcal{L} = \mathcal{L}_{\text{CE}} + \lambda_1 \mathcal{L}_{\text{SIoU}} + \lambda_2 \mathcal{L}_{\text{KL}}. \tag{1}$$

where $\mathcal{L}_{\text{CE}}$, $\mathcal{L}_{\text{SIoU}}$ and $\mathcal{L}_{\text{KL}}$ denote the cross-entropy loss, SIoU loss and KL divergence, respectively. The $\lambda$ values are used as weights to balance the contribution of each loss term.

**Inference.** During inference, the trajectory is initialized using the object bounding box in the first frame. The inter-frame sparsification method removes redundant and noisy tokens from templates, retaining and propagating these sparsification results through subsequent tracking processes to reduce computational complexity. The trajectory tokens are iteratively propagated into subsequent frames in an autoregressive manner. Unlike the training phase, since sparsification has already been performed on the templates, LayerNorm is applied to all tokens as usual.

## 4 Experiments

### 4.1 Implementation Details

The model is trained with 8 NVIDIA RTX A6000 GPUs. The inference speed is evaluated with NVIDIA TiTan Xp, Intel(R) Xeon(R) Gold 6230R CPU @ 3.00GHz, and Ascend 310B.

**Model Variants.** We trained five variants of FARTrack with different configurations as Table 2.

Table 1: State-of-the-art comparison on GOT-10k Huang et al. (2019), TrackingNet Muller et al. (2018) and LaSOT Fan et al. (2019). Best in **bold**, second best underlined.

| Methods | GPU FPS | CPU FPS | NPU FPS | GOT-10k AO(%) | $SR_{50}$(%) | $SR_{75}$(%) | TrackingNet AUC(%) | $P_{Norm}$(%) | P(%) | LaSOT AUC(%) | $P_{Norm}$(%) | P(%) |
|---|---|---|---|---|---|---|---|---|---|---|---|---|
| DiMP Bhat et al. (2019) | 68 | 22 | - | - | - | - | 74.0 | 80.1 | 70.6 | 56.9 | 65.0 | 56.7 |
| SiamFC++ Xu et al. (2020) | 76 | 28 | - | - | - | - | 75.4 | 80.0 | 68.7 | 54.4 | 62.3 | 54.7 |
| LightTrack Yan et al. (2021b) | 59 | 27 | - | 61.1 | 71.0 | 54.3 | 72.5 | 77.8 | 69.5 | 53.8 | 60.5 | 53.7 |
| TCTrack Cao et al. (2022) | 42 | 29 | - | 66.2 | 75.6 | 61.0 | 74.8 | 79.6 | 73.3 | 60.5 | 69.3 | 62.4 |
| FEAR Borsuk et al. (2022) | 123 | 28 | - | 61.9 | 72.2 | 52.5 | 70.2 | 80.8 | 71.5 | 53.5 | 59.7 | 54.5 |
| E.T. Track Blatter et al. (2023) | 33 | 16 | - | 56.6 | 64.6 | 42.5 | 72.5 | 77.8 | 69.5 | 59.1 | 66.8 | 60.1 |
| HiT-Tiny Kang et al. (2023) | 135 | 42 | 56 | 52.6 | 59.3 | 42.7 | 74.6 | 78.1 | 68.8 | 54.8 | 60.5 | 52.9 |
| HiT-Small Kang et al. (2023) | 121 | 35 | 47 | 62.6 | 71.2 | 54.4 | 77.7 | 81.9 | 73.1 | 60.5 | 68.3 | 61.5 |
| HiT-Base Kang et al. (2023) | 116 | 30 | 33 | 64.0 | 72.1 | 58.1 | 80.0 | 84.4 | 77.3 | 64.6 | 73.3 | **68.1** |
| MixformerV2 Cui et al. (2023) | 133 | 31 | 35 | 61.9 | 71.7 | 51.3 | 75.8 | 81.1 | 70.4 | 60.6 | 69.9 | 60.4 |
| LiteTrack-B4 Wei et al. (2024) | 195 | 29 | - | - | - | - | 79.9 | 84.9 | 76.6 | 62.5 | 72.1 | 65.7 |
| PromptVT Zhang et al. (2024) | 104 | 30 | - | 68.2 | 79.3 | 61.8 | 78 | 83.5 | 74.4 | 63.7 | **73.8** | 66.8 |
| SMAT Gopal & Amer (2024) | 135 | 40 | - | 64.5 | 74.7 | 57.8 | 78.6 | 84.2 | 75.6 | 61.7 | 71.1 | 64.6 |
| ECTTrack Xu et al. (2025) | 104 | 46 | - | 65.6 | 75 | 60.7 | 78.8 | 84.6 | 76.5 | 62.4 | 71.5 | 66.3 |
| CompressTracker Hong et al. (2025) | 207 | 48 | - | - | - | - | 78.2 | 83.3 | 74.8 | 60.4 | 68.5 | 61.5 |
| AsymTrack-T Zhu et al. (2025) | 145 | 55 | - | 62.3 | 71.3 | 54.7 | 76.2 | 80.9 | 71.6 | 60.8 | 68.7 | 61.2 |
| AsymTrack-S Zhu et al. (2025) | 136 | 48 | - | 65.5 | 74.8 | 58.9 | 77.9 | 82.2 | 74.0 | 62.8 | 71.2 | 64.8 |
| AsymTrack-B Zhu et al. (2025) | 135 | 32 | - | 67.7 | 76.6 | 61.4 | 80.0 | 84.5 | 77.4 | **64.7** | 73.0 | 67.8 |
| **FARTrack**pico | **343** | **121** | **101** | 62.8 | 72.6 | 50.9 | 75.6 | 81.3 | 70.5 | 58.6 | 67.1 | 59.6 |
| **FARTrack**nano | 210 | 77 | 61 | 69.9 | 81.2 | 61.4 | 79.1 | 84.5 | 75.6 | 61.3 | 69.7 | 64.1 |
| **FARTrack**tiny | 135 | 53 | 42 | **70.6** | **81.0** | **63.8** | **80.7** | **85.6** | **77.5** | 63.2 | 71.6 | 66.7 |

The tiny is a 15-layer encoder model. Nano distills the 15-layer encoder into 10 layers, and pico into 6 layers. Tiny matches AsymTrack-B on GPU while keeping AsymTrack-T CPU speed. Nano surpasses CompressTracker-OSTrack-2 Hong et al. (2025) in terms of key evaluation metrics on all datasets. Pico outperforms MixFormerV2-S Cui et al. (2023) by 0.9% in AO on GOT-10k Huang et al. (2019), showing FARTrack's efficiency.

**Training.** To conduct a fair comparison with mainstream trackers, we carried out Frame-level Pretraining on the COCO2017 Lin et al. (2014) dataset. Subsequently, we performed Task-Specific Self-Distillation Training to compress our model. Finally, we conducted Inter-frame Autoregressive Sparsification Training to further accelerate our model. The detailed training process can be found in the supplementary material.

## 4.2 MAIN RESULTS

We evaluated the performance of our proposed FARTrack_tiny, FARTrack_nano, and FARTrack_pico on several benchmarks, including GOT-10k Huang et al. (2019), TrackingNet Muller et al. (2018), LaSOT Fan et al. (2019), LaSOText Fan et al. (2020), NFS Kiani Galoogahi et al. (2017), UAV123 Mueller et al. (2016) and VastTrack Peng et al. (2024).

**GOT-10k Huang et al. (2019).** GOT-10k is a real world general object detection dataset. As shown in Table 1, FARTrack_tiny outperforms AsymTrack-B by 2.9% in AO score, achieving a GPU speed of 135 FPS and a CPU speed of 53 FPS. Furthermore, the most lightweight version, FARTrack_pico, outperforms MixFormerV2-S by 0.9% in AO, while delivering nearly three times the GPU speed and four times the CPU speed.

**TrackingNet Muller et al. (2018).** TrackingNet is a large-scale dataset featuring over 30,000 videos from diverse real world scenes. The evaluation on this extensive dataset highlights the efficiency and effectiveness of FARTrack. As illustrated in Table 1, FARTrack_nano achieves performance close to that of AsymTrack-B, the top-performing tracker on this benchmark, while running nearly twice as fast as AsymTrack-B on the GPU.

Table 2: Details of our FARTrack model variants

| Model | FARTrack$_{tiny}$ | FARTrack$_{nano}$ | FARTrack$_{pico}$ |
|---|---|---|---|
| Backbone | ViT-Tiny | ViT-Tiny | ViT-Tiny |
| Encoder Layers | 15 | 10 | 6 |
| Input Sizes | [112,224] | [112,224] | [112,224] |
| Templates | 5 | 5 | 5 |
| MACs (G) | 2.65 | 1.78 | 1.08 |
| Params (M) | 6.82 | 4.59 | 2.81 |

Table 3: Comparison on VastTrack benchmark.

| Methods | VastTrack | | |
|---|---|---|---|
| | AUC(%) | P$_{Norm}$(%) | P(%) |
| **FARTrack$_{tiny}$** | **35.2** | **36.5** | 32.3 |
| **FARTrack$_{nano}$** | 33.9 | 35.1 | 30.3 |
| **FARTrack$_{pico}$** | 30.3 | 31.0 | 25.7 |
| MixformerV2-B | 35.2 | 36.5 | **33.0** |
| DiMP | 29.9 | 31.7 | 25.7 |

**LaSOText Fan et al. (2020).** LaSOText is an extended subset of LaSOT, encompassing 150 additional videos from 15 new categories. As shown in Table 4, FARTrack$_{tiny}$ outperforms AsymTrack-B, with a 0.4% AUC improvement, demonstrating its effectiveness in small object tracking.

**NFS Kiani Galoogahi et al. (2017).** The NFS is a high-frame-rate benchmark dedicated to scenarios with fast-moving objects. As shown in Table 4, on the 30 FPS version of the NFS dataset, FARTrack$_{tiny}$ achieves outstanding performance in these challenging scenarios and attains the highest AUC score.

**UAV123 Mueller et al. (2016).** The UAV123 focuses on the unique challenges of UAV-based visual tracking. As shown in Table 4, FARTrack$_{tiny}$ delivers competitive performance on this dataset, outperforming AsymTrack-T by 1.2%. With comparable GPU and CPU inference speeds, it also surpasses AsymTrack-S by 0.2%.

Table 4: State-of-the-art comparison on more benchmarks.

| Methods | GPU | CPU | LaSOText | NFS | UAV123 |
|---|---|---|---|---|---|
| | FPS | FPS | AUC(%) | AUC(%) | AUC(%) |
| LightTrack | 59 | 27 | - | 55.3 | 62.5 |
| FEAR | 123 | 28 | - | 61.4 | - |
| E.T. Track | 33 | 16 | - | 59.0 | 62.3 |
| HiT-Tiny | 135 | 42 | 35.8 | 53.2 | 58.7 |
| HiT-Small | 121 | 35 | 40.4 | 61.8 | 63.3 |
| HiT-Base | 116 | 30 | 44.1 | 63.6 | 65.6 |
| MixformerV2 | 133 | 31 | 43.6 | - | 65.8 |
| LiteTrack-B4 | 195 | 29 | - | 63.4 | 66.4 |
| SMAT | 135 | 40 | - | 62.0 | 64.3 |
| ECTTrack | 104 | 46 | 39.9 | 61.1 | **67.0** |
| CompressTracker | 207 | 48 | 40.4 | - | 62.5 |
| AsymTrack-T | 145 | 55 | 42.5 | 63.3 | 64.6 |
| AsymTrack-S | 136 | 48 | 43.3 | 64.9 | 65.6 |
| AsymTrack-B | 135 | 32 | 44.6 | 64.4 | 66.5 |
| **FARTrack$_{pico}$** | **343** | **121** | 41.8 | 62.0 | 63.1 |
| **FARTrack$_{nano}$** | 210 | 77 | 43.8 | 65.1 | 62.6 |
| **FARTrack$_{tiny}$** | 135 | 53 | **45.0** | **66.9** | 65.8 |

**LaSOT Fan et al. (2019).** LaSOT is a large-scale benchmark designed to assess the robustness of long-term tracking. As demonstrated in Table 1, FARTrack$_{tiny}$ achieves AUC of 2.6% over MixFormerV2-S on this dataset. While maintaining a comparable running speed on the GPU, it nearly doubles the speed on the CPU. FARTrack$_{nano}$ matches AUC of MixFormerV2-S but surpasses it in both GPU and CPU running speed.

**VastTrack Peng et al. (2024).** VastTrack is a dataset aimed at advancing the development of more general visual tracking technology, covering 2,115 object categories and containing 50,610 video sequences. As shown in Table 3, on this dataset, FARTrack$_{tiny}$ achieves an AUC comparable to that of MixFormerV2-B, which fully highlights the robustness of FARTrack.

## 4.3 EXPERIMENTAL ANALYSES

We analyze the main properties of the FARTrack. For the following experimental studies, we follow the GOT-10k test protocol unless otherwise noted. Default settings are marked in gray .

**Distillation Strategy.** We compare layer-by-layer distillation with cross-layer distillation to validate our method. Cross-layer distillation uses an intermittent student-teacher layer correspondence, which aids knowledge transfer but introduces feature consistency issues. Considering the weaker representation capacity of ViT-Tiny, we also conduct ViT-Base-guided distillation to verify the superiority of layer-by-layer distillation.

As shown in Table 5, we design a manual layer reduction strategy following the Deep-to-Shallow distillation in MixformerV2, where *REMOVE_LAYERS* for 10-layer and 6-layer models are set to [0, 3, 6, 9, 12] and [0, 2, 4, 6], respectively. Removing model layers allows the remaining

Table 5: vs. Deep-to-Shallow Distillation Cui et al. (2023).

| Methods | Layer | AO | $SR_{50}$ | $SR_{75}$ |
|---|---|---|---|---|
| layer-by-layer | 10 | 69.9 | 81.2 | 61.4 |
| | 6 | 62.8 | 72.6 | 50.9 |
| deep-to-shallow | 10 | 67.8 | 78.0 | 60.6 |
| | 6 | 61.9 | 70.9 | 50.4 |

Table 6: Sparsification Comparison.

| run-time | sequence-time | MACs | Params | CPU | GPU | AO |
|---|---|---|---|---|---|---|
| | | 2.99G | 6.82M | 49 | 128 | 70.0 |
| ✓ | | 3.14G | 7.21M | 36 | 114 | 69.5 |
| | ✓ | **2.65G** | **6.82M** | **53** | **135** | **70.6** |

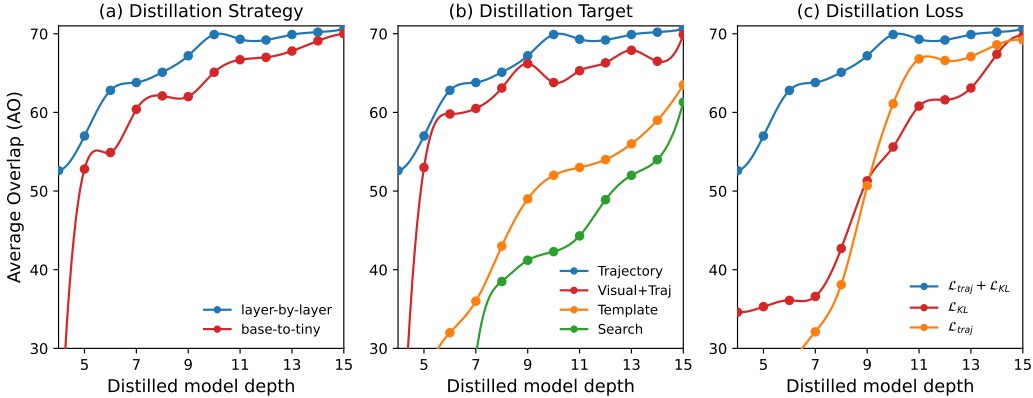

Figure 4: Layer-by-layer distillation accuracy curve.

layers to perform sequential inter-layer matching. However, manual layer assignment fails to ensure reasonable alignment, causing semantic mismatch in cross-layer distillation. This mismatch breaks feature alignment and leads to severe accuracy drop in deeper layers. In contrast, our method preserves feature consistency and effectively alleviates semantic mismatch.

As illustrated in Figure 4(a), FARTrack with ViT-Base shares an identical hierarchical structure with FARTrack$_{tiny}$. We distill each layer of ViT-Tiny using trajectory features from corresponding ViT-Base layers. Although the 15-layer base-to-tiny distilled model performs well, forcing Tiny to mimic Base's outputs induces severe accuracy degradation in deeper layers (10–14) and progressive decline in shallower layers due to representational capability mismatch. In contrast, our method preserves feature consistency and information density, maintaining nearly identical accuracy in layers 10–15 while constraining shallow-layer accuracy loss within an acceptable range.

**Distillation Target.** To verify the necessity of distilling the trajectory sequence, we conducted an exploratory experiment, as shown in Figure 4(b). Experiments reveal that directly distilling the search/template or jointly distilling visual features disrupts the hierarchical structure, causing accuracy degradation across all layers. In contrast, distilling the trajectory sequence minimally impacts the feature extraction hierarchy. Its autoregressive properties enhance knowledge propagation from deep to shallow layers, maintaining nearly unchanged accuracy in layers 10-15 while preventing significant performance drops in shallow layers.

**Distillation Loss.** Experiments (Figure 4(c)) demonstrate that combining trajectory sequence loss ($\mathcal{L}_{traj} = \mathcal{L}_{CE} + \mathcal{L}_{SIoU}$) and KL divergence loss ($\mathcal{L}_{KL}$) effectively preserves accuracy in deep layers (*e.g.*, layers 10-15) while controlling performance degradation in shallow layers. Removing trajectory sequence loss causes significant feature degradation and performance drop, proving its critical role in preserving temporal information. In contrast, omitting KL divergence loss triggers rapid feature collapse in shallow layers, leading to catastrophic degradation of tracking performance, highlighting its indispensability for maintaining object tracking capability.

**Sparsification Comparison.** The runtime sparsification introduces extra computational processes during the inference, while our sequence-level sparsification avoids this situation. To support this, we conducted experiments on runtime sparsification and sequence-level sparsification respectively based on the base model, and compared the final results, as shown in Table 6.

Table 7: Token Retention Ratio.

| Ratio | MACs | CPU | GPU | AO | $SR_{50}$ | $SR_{75}$ |
|---|---|---|---|---|---|---|
| 100% | 2.99G | 49 | 128 | 70.0 | 80.0 | 64.4 |
| **75%** | **2.65G** | **53** | **135** | **70.6** | **81.0** | **63.8** |
| 50% | 2.35G | 56 | 139 | 68.3 | 78.2 | 61.8 |
| 25% | 2.01G | 58 | 140 | 67.3 | 78.3 | 59.8 |
| 10% | 1.82G | 63 | 141 | 62.3 | 74.1 | 53.3 |

Table 8: Attention Map Sampling Method.

| $S_{1\times1}$ | $S_{3\times3}$ | $S_{all}$ | $Z$ | $C$ | AO | $SR_{50}$ | $SR_{75}$ |
|---|---|---|---|---|---|---|---|
| | | | | | 70.1 | 80.6 | 63.3 |
| ✓ | | | | | 69.5 | 80.4 | 62.4 |
| | ✓ | | | | 70.0 | 80.4 | 62.9 |
| | | ✓ | | | 68.6 | 79.1 | 61.7 |
| | ✓ | | ✓ | | 69.4 | 79.7 | 62.1 |
| | ✓ | | | ✓ | **70.6** | **81.0** | **63.8** |

Table 9: Comparison with ARTrack Wei et al. (2023a) and ARTrackV2 Bai et al. (2024).

| Methods | Self-distillation | Sparsification | MACs | Params | GPU | CPU | GOT-10k | | |
|---|---|---|---|---|---|---|---|---|---|
| | | | G | M | FPS | FPS | AO(%) | $SR_{50}$(%) | $SR_{75}$(%) |
| ARTrack_tiny(6) | | | 2.39 | 13.56 | 68 | 34 | 59.9 | 69.6 | 50.4 |
| ARTrackV2_tiny(6) | | | 2.10 | 8.31 | 110 | 49 | 60.6 | 70.9 | 45.7 |
| FARTrack$_{pico}$(6) | | ✓ | 1.08 | 2.81 | 343 | 121 | 60.6 | 69.5 | 46.3 |
| FARTrack$_{pico}$(6) | ✓ | | 1.25 | 2.81 | 266 | 101 | 61.8 | 71.5 | 50.0 |
| **FARTrack$_{pico}$(6)** | ✓ | ✓ | **1.08** | **2.81** | **343** | **121** | **62.8** | **72.6** | **50.9** |

Runtime sparsification processes template tokens during each forward propagation in inference, introducing extra computational overhead that increases MACs from 2.99G to 3.14G and Params from 6.82M to 7.21M. This redundant computation reduces CPU and GPU speeds by 26.5% and 10.9% respectively. In contrast, our sequence-level sparsification leverages intermediate results for decision-making without introducing additional computations, while propagating sparsification outcomes to subsequent frames autoregressively to eliminate redundant processing. Consequently, MACs are reduced to 2.65G with improved inference speeds on both CPU and GPU.

**Token Retention Ratio.** Token retention ratio in inter-frame autoregressive sparsification affects both performance and efficiency. As Table 7 shows, reducing the ratio from 100% to 75% decreased MACs by 11.4% (2.99G to 2.65G) while achieving peak AO (70.6%). This indicates significant redundancy in target templates—removing 25% background tokens preserves temporal modeling and improves accuracy. Further reduction to 25% caused 3.3% AO drop (67.3%), demonstrating excessive removal harms tracking robustness.

**Attention Map Sampling Method.** We analyze different sampling strategies for inter-frame autoregressive sparsification in Table 8. $S_{1\times1}$ sparsifies the central $1 \times 1$ feature, $S_{3\times3}$ uses a $3 \times 3$ central region, $S_{all}$ covers the entire search area. $Z$ denotes template self-attention, while $C$ refers to command-template attention.

$S_{3\times3}$ avoids target exclusion from center shift or hollow structures and reduces background overfocus versus $S_{all}$, improving accuracy. Combining $S$ and $Z$ underperforms due to self-overfocus in $Z$. Instead, integrating $S$ and $C$ enables effective template sparsification: $S$ separates foreground and background coarsely, while $C$ refines edge features. This reduces redundancy and improves accuracy and efficiency.

**Independent Performance Gains of Each Module.** To analyze the performance gains from each module, we perform supplementary comparisons with ARTrack and ARTrackV2. ARTrack, ARTrackV2, and FARTrack$_{pico}$ all employ the ViT-Tiny architecture with 6 encoder layers, and share the same training settings and datasets as FARTrack.

As shown in Table 9, it strongly validates the effectiveness of the self-distillation module and the sparsification module. The self-distillation module, which enables the middle layers of the network to learn knowledge from the deep layers, improves the AO by 1.2% compared with ARTrackV2_tiny (6). The sparsification module, which retains useful foreground feature tokens while eliminating redundant background feature tokens, boosts the AO by 1.0% (from 61.8 to 62.8).

**Layer-wise cross-attention visualization.** As shown in Figure 5, cross-attention between trajectory sequences and search regions evolves across layers: shallow layers (0–4) capture edges and back-

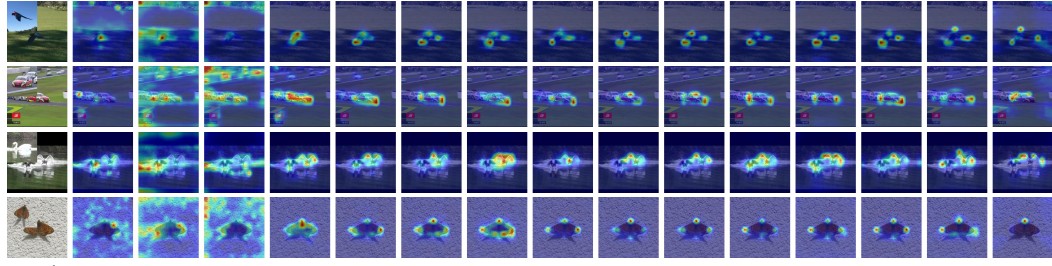

search | layer 0 | layer 1 | layer 2 | layer 3 | layer 4 | layer 5 | layer 6 | layer 7 | layer 8 | layer 9 | layer 10 | layer 11 | layer 12 | layer 13 | layer 14

Figure 5: **Layer-wise cross-attention visualization.** search: Search region and template. layer 0-14: Trajectory sequences to search cross-attention maps at each layer.

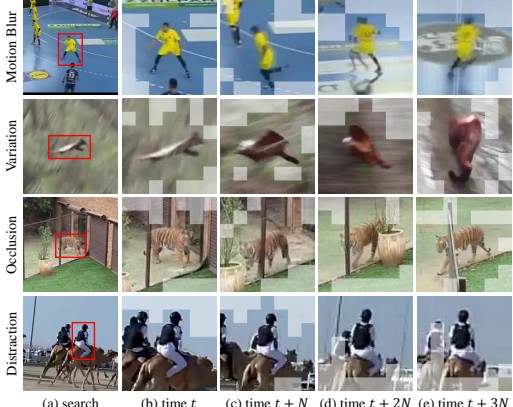

(a) search | (b) time $t$ | (c) time $t + N$ | (d) time $t + 2N$ | (e) time $t + 3N$

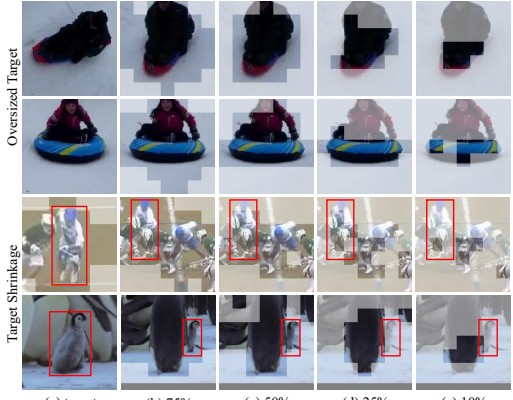

(a) target | (b) 75% | (c) 50% | (d) 25% | (e) 10%

Figure 6: **Template sparsification visualization.** (a): Search region. The red boxes denote the ground truth. (a)-(e): Templates sampled at a fixed interval of $N$ and sparsified with a 75% token retention ratio.

Figure 7: **Template retention visualization.** (a): The initial template of the video sequence. The red boxes denote the ground truth. (b)–(e): The template at time step $t$, sparsified with different token retention ratios.

ground, while deeper ones (9–14) transition toward target contours. Hierarchical feature learning is retained via distillation, maintaining consistent trajectory propagation and boosting accuracy, especially in intermediate layers (*e.g.*, 5, 9).

**Template sparsification visualization.** Figure 6 shows that our method retains critical tokens and removes redundancy under motion blur, appearance change, and occlusion. Unlike frame-wise sparsification, which often fails due to single-frame errors, our inter-frame autoregressive approach uses multi-template complementarity and temporal modeling to track dynamic targets and preserve structure even with inconsistent cues.

**Template retention visualization.** Figure 7 visualizes retained tokens at different retention ratios. For oversized targets, low ratios ($<50\%$) lose essential features; for shrinking ones, they harm deformation representation and cause misclassification. Thus, we set a 75% retention ratio to preserve sufficient target information while reducing redundancy, balancing accuracy and efficiency.

# 5 CONCLUSION

We propose FARTrack, a fast and high-performance multi-template autoregressive tracking framework. It integrates task-specific self-distillation and inter-frame autoregressive sparsification. While slightly behind top-performing methods Bai et al. (2024); Wei et al. (2023a), it excels in speed-performance balance, especially in speed. Our distillation preserves temporal information of trajectory sequences via layer-wise task-specific tokens distillation, avoiding suboptimal manual layer assignments. The sparsification method propagates multi-template sparsification results autoregressively, achieving temporally-global optimality without extra cost. FARTrack performs well across GPU, CPU, and NPU platforms, offering an efficient solution for practical deployment.

## ACKNOWLEDGMENTS

This work was support by the National Natural Science Foundation of China No. 62572385, the Fundamental Research Funds for the Central Universities No. xxj032023020, and CAAI-CANN Open Fund, developed on OpenI Community.

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

APPENDIX

## A    TRAINING DETAILS

In the supplementary material, we have supplemented the experiments section. We have sequentially presented the detailed training processes of Frame-level Pretraining, Task-Specific Self-Distillation Training, and Inter-frame Autoregressive Sparsification Training.

**Frame-level Pretraining.**    To fairly compare with mainstream trackers, we introduce the COCO2017 Lin et al. (2014) dataset, which is commonly used in template matching training paradigms, and apply frame-level pretraining to the AR(0) model. Similar to DiMP Bhat et al. (2019) and STARK Yan et al. (2021a), the AR(0) model uses the same data augmentations as OS-Track, including horizontal flip and brightness jittering. The model is optimized using AdamW with a weight decay of $1 \times 10^{-4}$. The learning rate for the backbone is set to $4 \times 10^{-4}$, and $4 \times 10^{-3}$ for other parameters. The AR(0) model is trained for 500 epochs, with 76,800 template-search frame pairs sampled per epoch. The learning rate is reduced by 10% at the 400th epoch.

**Task-Specific Self-Distillation Training.** This phase uses the same datasets and data augmentations as the AR(0) phase but introduces additional KL divergence loss ($\mathcal{L}_{\mathrm{KL}}$) and trajectory sequence loss ($\mathcal{L}_{\mathrm{traj}} = \mathcal{L}_{\mathrm{CE}} + \mathcal{L}_{\mathrm{SIoU}}$) for each layer. The model is optimized using AdamW with a weight decay of $1 \times 10^{-4}$. The learning rate for the backbone is set to $4 \times 10^{-5}$, and $4 \times 10^{-4}$ for other parameters. The distillation model is trained for 300 epochs, with 76,800 template-search frame pairs randomly sampled per epoch. This process generates multiple versions of the distilled model, producing models at different layers from a single distillation.

**Inter-frame Autoregressive Sparsification Training.** Unlike traditional per-frame template matching, this method trains FARTrack directly on continuous video sequences without applying data augmentations. The model is optimized using AdamW with a weight decay of $5 \times 10^{-2}$. The learning rate for the backbone is set to $4 \times 10^{-7}$ and $4 \times 10^{-6}$ for other parameters. The training process consists of 20 epochs, with 1,000 video slices randomly sampled from continuous video per epoch. Due to GPU memory limitations, each slice contains 32 frames.

## B    TEMPLATE QUANTITY

In this section, an ablation experiment is conducted on the number of templates. The number of templates not only affects the operation efficiency of the model but also impacts the tracking accuracy. Therefore, it is essential to explore this aspect.

Table 10: Impact of Templates on Efficiency and Performance.

| Template Count | MACs | Params | CPU | GPU | AO | SR$_{50}$ | SR$_{75}$ |
|---|---|---|---|---|---|---|---|
| 1 | 1.70G | 6.82M | 64 | 141 | 66.4 | 77.0 | 58.0 |
| 3 | 2.17G | 6.82M | 55 | 139 | 68.1 | 78.6 | 60.9 |
| **5** | **2.65G** | **6.82M** | **53** | **135** | **70.6** | **81.0** | **63.8** |
| 7 | 3.13G | 6.82M | 48 | 124 | 69.6 | 80.3 | 62.5 |
| 9 | 3.61G | 6.82M | 45 | 115 | 70.0 | 80.7 | 62.3 |

As shown in Table 10, when the number of templates gradually increases from 1 to 5, the AO gradually rises from 66.4% to 70.6%, and the MACs increases synchronously by 35.8% (from 1.70G to 2.65G). During this stage, adding every two templates can increase the AO by approximately 2.1%, which verifies that the multi-template mechanism can effectively aggregate the appearance changes of the target and achieve an accurate representation of the target's dynamic appearance. However, when the number of templates exceeds 5 (*e.g.*, 7-9), the AO instead drops to the range of 69.6%-70.0%. This indicates that redundant templates lead to the dispersion of attention weights and interfere with the extraction of key features.

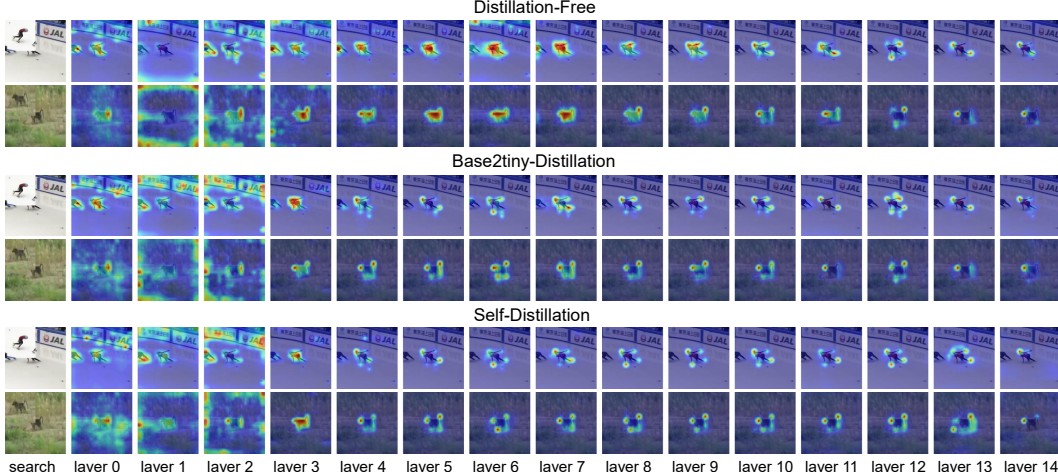

Figure 8: **Layer-wise cross-attention visualization Comparison.** search: Search region and template. layer 0-14: Trajectory sequences to search cross-attention maps at each layer.

Finally, we choose to use 5 templates because it achieves the optimal efficiency at the critical point of the accuracy peak.

## C  TEMPLATE UPDATE STRATEGY

In this section, an ablation experiment is conducted on the template update strategy. Different test benchmarks have an obvious correlation with different template update strategies. Therefore, it is necessary to experiment with different update strategies.

Table 11: Template Update Strategy Comparison.

| Update Strategy | GOT-10k | | | LaSOT | | |
|---|---|---|---|---|---|---|
| | AO(%) | $SR_{50}$(%) | $SR_{75}$(%) | AUC(%) | $P_{Norm}$(%) | P(%) |
| Linear | 70.9 | 81.6 | 63.4 | 62.5 | 70.2 | 65.3 |
| **Exponential Decay** | **70.6** | **81.0** | **63.8** | **63.2** | **71.6** | **66.7** |
| Logarithmic Increasing | 69.6 | 80.1 | 62.8 | 61.6 | 68.7 | 65.4 |
| Equal-interval | 69.5 | 79.4 | 63.1 | 62.1 | 69.3 | 65.5 |

As shown in Table 11, linear sampling achieves relatively mediocre values among all the update strategies. However, since it always samples all templates linearly, it is not sensitive to the length of the target sequence, and thus it has good performance on different datasets. Exponential decay sampling samples more frames closer to the current frame. This is more friendly to long sequences such as the LaSOT benchmark and also performs well in short video sequences like GOT-10k. Logarithmic increasing sampling samples more frames that are farther from the current frame, which makes it perform poorly in long video sequences such as LaSOT. Equal-interval sampling performs poorly both in GOT-10k and LaSOT. This is because equal-interval sampling may fail to sample the initial static template. And if there is a long period of tracking failure, such as the target disappearing or being occluded, it is very likely that all templates will become invalid, resulting in a decrease in accuracy.

## D  INTER-LAYER ATTENTION VISUALIZATION COMPARISON

In this section, we conduct ablation experiments on the inter-layer attention of different distillation methods. Given the evident correlation between the distribution patterns of inter-layer attention and various distillation approaches, experimental validation is therefore deemed necessary.

For the layer-wise cross-attention visualized in Figure 8, the inter-layer attention distributions under the distillation-free setting exhibit diversity with distinct differences across various layers. Compared

with self-distillation, the inter-layer attention distributions in the base2tiny distillation setting are more scattered. In contrast, the deep-layer attention distributions after trajectory self-distillation demonstrate consistency — a characteristic that confirms the effectiveness of the self-distillation mechanism in transferring deep-layer knowledge to middle layers (*e.g.*, layers 5–14). Thus, compared with other configurations, self-distillation enables more coherent inter-layer feature alignment.

# E  COMPARATIVE ANALYSIS OF DISTILLATION VS. SCRATCH-TRAINED MODELS WITH DIFFERENT DEPTHS

We have supplemented comparative experiments between 10-layer or 6-layer models trained from scratch and our distilled versions to clarify whether distillation provides benefits beyond simply using fewer layers (see Table 12).

Table 12: Performance Comparison of Different FARTrack Variants and Their Corresponding Scratch-Trained Models.

| Methods | GOT-10k | | | TrackingNet | | | LaSOT | | |
|---|---|---|---|---|---|---|---|---|---|
| | AO(%) | $SR_{50}$(%) | $SR_{75}$(%) | AUC(%) | $P_{Norm}$(%) | P(%) | AUC(%) | $P_{Norm}$(%) | P(%) |
| FARTrack_tiny | 70.6 | 81.0 | 63.8 | 80.7 | 85.6 | 77.5 | 63.2 | 71.6 | 66.7 |
| **FARTrack_nano(10)** | 69.9 | 81.2 | 61.4 | 79.1 | 84.5 | 75.6 | 61.3 | 69.7 | 64.1 |
| FARTrack_scratch(10) | 67.1 | 77.4 | 59.9 | 77.9 | 83.1 | 73.6 | 60.8 | 69.1 | 63.3 |
| **FARTrack_pico(6)** | 62.8 | 72.6 | 50.9 | 75.6 | 81.3 | 70.5 | 58.6 | 67.1 | 59.6 |
| FARTrack_scratch(6) | 60.6 | 69.5 | 46.3 | 73.3 | 78.4 | 67.1 | 56.8 | 64.1 | 55.7 |

The results confirm that distillation offers advantages far beyond simply using fewer layers:

*(i)* **10-layer or 6-layer models trained from scratch:** Suffer severe performance degradation (e.g., FARTrack_scratch(10) exhibits a 3.5% drop in AO compared to FARTrack_tiny). This is because such models lack the learning capacity of deep networks and fail to capture complex visual/temporal features required for tracking.

*(ii)* **Our distilled versions:** Transfer deep-layer knowledge to middle layers via self-distillation, retaining higher performance (FARTrack_nano(10) only sees a 0.7% AO drop compared to FARTrack_tiny) while achieving lower training costs than 10-layer or 6-layer models trained from scratch.

# F  TRAINING COST COMPARISON: CROSS-LAYER VS. TASK-SPECIFIC SELF-DISTILLATION

We have supplemented the training cost comparison between cross-layer distillation (Deep-to-Shallow) Cui et al. (2023) and our task-specific self-distillation, with details as follows.

Both methods were trained on 8 NVIDIA RTX A6000 GPUs:

*(i)* Deep-to-Shallow distillation: Stage 1 (15-to-10 layers) takes about 40 hours; Stage 2 (10-to-6 layers) takes about 38 hours (two-stage sequential training required).

*(ii)* Task-specific self-distillation: Takes about 36 hours total, with one-time training enabling direct utilization of multiple model versions.

This comparison fully demonstrates the training efficiency advantage of our proposed method.

# G  THE USE OF LARGE LANGUAGE MODELS (LLMS)

We only used LLMs minimally to aid or polish writing. For instance, when describing a concept, we might leverage LLMs to ensure terminological precision, enhance logical coherence, or optimize the academic tone of the exposition.

