# OpenReview forum: "FARTrack: Fast Autoregressive Visual Tracking with High Performance"
_ICLR.cc/2026/Conference — ICLR 2026 Poster_

### Official Review · Reviewer_wSp9 · 2025-10-30

**Soundness:** 4
**Presentation:** 4
**Contribution:** 4
**Rating:** 8
**Confidence:** 5

**Summary:**

This paper aims to make the Autoregressive Visual Tracking method more lightweight for deployment. It presents two key technical contributions: the first is self-distillation for multiple templates, and the second is token sparsification for the tokens being processed. The self-distillation module learns trajectory features layer by layer, enabling lightweight compression and avoiding the manual layer matching required in conventional cross-layer distillation. Meanwhile, the inter-frame sparsification module filters redundant background information from the template frames at the sequence level and propagates the sparse results in an autoregressive manner, thereby avoiding additional computational overhead. The experimental results are robust and comprehensive. The comparisons to the efficient trackers show that the method is effective and achieves the SOTA results in three tracking benchmarks. The author also presents many visualization results and ablation studies to show the effectiveness of the method.

**Strengths:**

1.FARTrack compresses the backbone via: 1) task-specific self-distillation method: each layer teaches the one below using the trajectory-token stream, 2) inter-frame autoregressive sparsification.
2.The experimental results are promising, demonstrating strong speed and accuracy in the efficient trackers.
3.The motivation is good. The autoregressive tracking methods need specific lightweight methods, especially considering the temporal domain.
4.The method is straightforward and exceptionally easy to understand. It has the potential to be applied to more autocratic autoregressive model pipelines.

**Weaknesses:**

1.The writing could be improved. For example, the terminology used in the paper is not precise enough. The word "performance" tends to describe the efficiency of the model, and using "accuracy" would more accurately depict the tracking effect of the model.
2.Too few models are compared on the VastTrack benchmark, and more models should be added for comparison.
3.It is suggested to supplement the comparison of training costs between cross-layer distillation and task-specific self-distillation to more fully demonstrate the advantages of the proposed method.

**Questions:**

1.In Figure 2, why is the ratio of the model’s speed to its performance used to represent the circle’s size instead of FLOPs?
2.Is there any basis for the setting of the number of templates in Table 2?
3.What is the basis for the REMOVE_LAYERS settings [0, 3, 6, 9, 12] and [0, 2, 4, 6] in the Deep-to-Shallow distillation?

---

> ### Author Response · Authors · 2025-11-20
> **Responses to Reviewer wSp9  (Part1 of 1)**
>
> > $\textbf{Weakness1:}$ The writing could be improved. For example, the terminology used in the paper is not precise enough. The word "performance" tends to describe the efficiency of the model, and using "accuracy" would more accurately depict the tracking effect of the model.
>
> Thank you for your valuable suggestion! We agree that "accuracy" is a more focused term for describing tracking effectiveness and have made appropriate revisions to the manuscript accordingly.
>
>
>
> ---
>
> > $\textbf{Weakness2:}$ Too few models are compared on the VastTrack benchmark, and more models should be added for comparison.
>
> Thank you for your valuable suggestion. We fully acknowledge your concern about the limited number of compared models on the VastTrack[1] benchmark. As noted, VastTrack’s official supported tracker list currently only provides two efficient trackers (MixformerV2-B and DiMP), which has constrained our ability to include more models for comparison. Due to time constraints for the discussion, we are unable to conduct additional experiments for expanding the comparison set. We will further enrich the comparative models on the VastTrack benchmark in future to make the evaluation more comprehensive.
>
> [1] Peng et al. VastTrack: Vast Category Visual Object Tracking. 2024.
>
>
>
> ---
>
> > $\textbf{Weakness3:}$ It is suggested to supplement the comparison of training costs between cross-layer distillation and task-specific self-distillation to more fully demonstrate the advantages of the proposed method.
>
> Thank you for this valuable suggestion. We have supplemented the training cost comparison between cross-layer distillation (Deep-to-Shallow) and our task-specific self-distillation, with details as follows:
>
> Both methods were trained on 8 NVIDIA RTX A6000 GPUs:
>
> 1. Deep-to-Shallow distillation: Stage 1 (15-to-10 layers) takes ~40 hours; Stage 2 (10-to-6 layers) takes ~38 hours (two-stage sequential training required).
> 2. Task-specific self-distillation: Takes ~36 hours total, with one-time training enabling direct utilization of multiple model versions.
>
> This comparison fully demonstrates the training efficiency advantage of our proposed method.
>
>
>
> ---
>
> > $\textbf{Question1:}$ In Figure 2, why is the ratio of the model's speed to its performance used to represent the circle's size instead of FLOPs?
>
> Thank you for your insightful question. We chose the ratio of model speed to performance for the circle size in Figure 2 for two key reasons:
>
> 1. The horizontal axis already fully reflects the model’s efficiency, so using this ratio better demonstrates the comprehensive balance between speed and performance via circle size.
> 2. Using FLOPs to determine circle size results in poor graph visibility, making it difficult to distinguish differences between models.
>
>
>
> ---
>
> > $\textbf{Question2:}$ Is there any basis for the setting of the number of templates in Table 2?
>
> Thank you for your question. The template count setting in Table 2 is determined by our ablation experiments (detailed in Appendix B, as follow Table), which explore the trade-off between efficiency and performance:
>
> | Template Count | MACs  | Params | CPU  | GPU  | AO              | SR₅₀            | SR₇₅            |
> | -------------- | ----- | ------ | ---- | ---- | --------------- | --------------- | --------------- |
> | 1              | 1.70G | 6.82M  | 64   | 141  | 66.4            | 77.0            | 58.0            |
> | 3              | 2.17G | 6.82M  | 55   | 139  | 68.1            | 78.6            | 60.9            |
> | 5              | 2.65G | 6.82M  | 53   | 135  | $\mathbf{70.6}$ | $\mathbf{81.0}$ | $\mathbf{63.8}$ |
> | 7              | 3.13G | 6.82M  | 48   | 124  | 69.6            | 80.3            | 62.5            |
> | 9              | 3.61G | 6.82M  | 45   | 115  | 70.0            | 80.7            | 62.3            |
>
> As shown, 5 templates achieve the optimal balance: AO peaks at 70.6% (with moderate MACs growth), while more templates ($\ge$7) cause attention dispersion and performance degradation.
>
>
>
> ---
>
> > $\textbf{Question3:}$ What is the basis for the REMOVE_LAYERS settings [0, 3, 6, 9, 12] and [0, 2, 4, 6] in the Deep-to-Shallow distillation?
>
> Thank you for your question. The REMOVE_LAYERS settings in our Deep-to-Shallow distillation are strictly aligned with the configuration of MixFormerV2’s Deep-to-Shallow distillation[1], adapted to our model’s layer count:
>
> - Stage 1 (15-to-10 layers): REMOVE_LAYERS = [0, 3, 6, 9, 12] (consistent with MixFormerV2’s stage1 logic of [0, 3, 6, 9] for 12-to-8 layers).
> - Stage 2 (10-to-6 layers): REMOVE_LAYERS = [0, 2, 4, 6] (directly adopting MixFormerV2’s stage2 setting for 8-to-4 layers).
>
> This ensures consistency with established distillation protocols for fair comparison.
>
> [1] Cui et al. MixFormerV2: Efficient Fully Transformer Tracking. 2023.

---

### Official Review · Reviewer_jdqf · 2025-10-30

**Soundness:** 3
**Presentation:** 3
**Contribution:** 3
**Rating:** 4
**Confidence:** 5

**Summary:**

This paper proposes a fast autoregressive visual tracking framework called FARTrack, aiming to achieve both high accuracy and real-time speed. Its main innovations include Task-Specific Self-Distillation and Inter-Frame Autoregressive Sparsification. In experiments, FARTrack achieved 70.6% AO performance on the GOT-10k dataset, reaching 135 FPS on GPU and maintaining 121 FPS on CPU. It significantly outperforms several state-of-the-art trackers, including AsymTrack, MixFormerV2, and HiT, in terms of speed and performance balance.

**Strengths:**

1.The paper is well written.

2.Improve the efficiency of tracker is a good direction in visual object tracking, which makes progress in this feild.

3.Use Inter-frame Autoregressive Sparsification sounds new.

**Weaknesses:**

1.Self-Distillation is not new. The concept has been proposed by diffrenet field.

2.The comparison methods are insufficient — most baselines are from earlier years, with no inclusion of 2024 works and only one paper from 2025. The paper should include more recent state-of-the-art methods for a fair and comprehensive comparison.

3.No MACs and Parameters analysis with sota methods.

**Questions:**

Can you provide more sota methods resutls? And compare the performance and complexity.

---

> ### Author Response · Authors · 2025-11-20
> **Responses to Reviewer jdqf (Part1 of 2)**
>
> > $\textbf{Weakness1:}$ Self-Distillation is not new. The concept has been proposed by different field.
>
> Thank you very much for this comment. We fully acknowledge that the concept of self-distillation is not new in other fields, but our core contribution lies in its task-driven redesign tailored specifically for visual tracking—a non-trivial adaptation that addresses the unique requirements of the tracking task.
>
> - $\textbf{Task-Specific Self-Distillation:}$ We are the first to introduce self-distillation into visual tracking, with trajectory sequences rather than visual features as the distillation target. Previous distillation methods have focused solely on visual features, overlooking the importance of temporal coherence—a property that has already been verified in prior autoregressive trackers (e.g., ARTrack, ARTrackV2), where trajectory tokens with temporal coherence inherently enhance temporal modeling capability and tracking robustness. Thus, our design retains temporal consistency and outperforms visual feature-based distillation approaches (Figure 4(b)).
>
> In essence, our contribution is not the invention of self-distillation itself, but its customization to tracking’s core demand of temporal modeling—a targeted adaptation that cannot be achieved by simply applying existing self-distillation frameworks to visual tracking.

---

> ### Author Response · Authors · 2025-11-20
> **Responses to Reviewer jdqf (Part2 of 2)**
>
> > $\textbf{Weakness2}$, $\textbf{Weakness3}$ and $\textbf{Question1:}$ Can you provide more sota methods results? And compare the performance and complexity.
>
> Thank you for your valuable suggestion. We have supplemented the performance and complexity comparisons with the latest state-of-the-art (SOTA) methods published in 2024 and 2025. (*) denotes results on GOT-10k obtained following the official one-shot protocol. The speeds of both GPU and CPU were tested on the same hardware device. Some trackers lack official speed testing code, with their speed performance and hardware specifications unreported in the papers.
>
> | Methods                      | GPU  | CPU  | MACs  | Params | NFS    |       | GOT-10k |         |        | TrackingNet |      |        | LaSOT    |      |
> | ---------------------------- | ---- | ---- | ----- | ------ | ------ | ----- | ------- | ------- | ------ | ----------- | ---- | ------ | -------- | ---- |
> |                              | FPS  | FPS  | G     | M      | AUC(%) | AO(%) | SR50(%) | SR75(%) | AUC(%) | Pnorm(%)    | P(%) | AUC(%) | Pnorm(%) | P(%) |
> | ToS-Tiny[1]                  | —    | —    | 0.87  | 4.07   | 64.3   | 66.8  | 77.5    | 59.9    | 78.8   | 84.1        | 75.2 | 63.0   | 72.7     | 65.7 |
> | SiamGAT*[2]                  | —    | —    | —     | —      | —      | 67.1  | 78.7    | 58.9    | 76.9   | 82.4        | 71.9 | 59.5   | 69.0     | 61.2 |
> | RTDiMP[3]                    | —    | —    | —     | —      | 66.2   | 68.2  | 79.9    | 58.8    | 78.1   | 82.9        | 72.8 | —      | —        | —    |
> | LiteTrack-B4(*)[4]           | 195  | 29   | 6.78  | 26.18  | 63.4   | 65.2  | 74.7    | 57.7    | 79.8   | 84.9        | 76.6 | 62.5   | 72.1     | 65.7 |
> | PromptVT[5]                  | 104  | 30   | 2.90  | 3.00   | —      | 68.2  | 79.3    | 61.8    | 78.0   | 83.5        | 74.4 | 63.7   | 73.8     | 66.8 |
> | SMAT[6]                      | 135  | 40   | —     | 3.78   | 62.0   | 64.5  | 74.7    | 57.8    | 78.6   | 84.2        | 75.6 | 61.7   | 71.1     | 64.6 |
> | ECTTrack[7]                  | 104  | 46   | —     | —      | 61.1   | 65.6  | 75.0    | 60.7    | 78.8   | 84.6        | 76.5 | 62.4   | 71.5     | 66.3 |
> | CompressTracker-OSTrack-2[8] | 207  | 48   | 6.38  | 21.24  | —      | —     | —       | —       | 78.2   | 83.3        | 74.8 | 60.4   | 68.5     | 61.5 |
> | SSTrack(*)[9]                | 62   | 18   | 32.55 | 92.12  | —      | 67.1  | 76.6    | 59.1    | 80.1   | 86.7        | 78.9 | 64.8   | 75.2     | 69.7 |
> | FARTrack_tiny                | 135  | 53   | 2.65  | 6.82   | 66.9   | 70.6  | 81.0    | 63.8    | 80.7   | 85.6        | 77.5 | 63.2   | 71.6     | 66.7 |
> | FARTrack_nano                | 210  | 77   | 1.78  | 4.59   | 65.1   | 69.9  | 81.2    | 61.4    | 79.1   | 84.5        | 75.6 | 61.3   | 69.7     | 64.1 |
> | FARTrack_pico                | 343  | 121  | 1.08  | 2.81   | 62.0   | 62.8  | 72.6    | 50.9    | 75.6   | 81.3        | 70.5 | 58.6   | 67.1     | 59.6 |
>
> As can be seen from the table results, our FARTrack achieves state-of-the-art (SOTA) performance on the NFS/GOT-10k/TrackingNet benchmark and competitive accuracy on the LaSOT benchmarks. On the LaSOT benchmark, the AUC of SSTrack is 1.6% higher than that of our tiny model (64.8% > 63.2%). This is attributed to the fact that its parameter count is is much larger than that of ours ($\mathbf{92.12M}$ > $\mathbf{6.82M}$), thus endowing SSTrack with stronger robustness in long-term temporal tracking. However, under the condition of comparable parameter counts, our model should deliver higher accuracy.
>
> [1] Zong et al. Enhancing the Two-Stream Framework for Efficient Visual Tracking. IEEE TIP'2025
>
> [2] Shao et al. Graph Attention Network for Context-Aware Visual Tracking. IEEE TNNLS'2024
>
> [3] Wang et al. Dynamic Region-Aware Transformer Backbone Network for Visual Tracking. article'2024
>
> [4] Wei et al. LiteTrack: Layer Pruning with Asynchronous Feature Extraction for Lightweight and Efficient Visual Tracking. ICRA'2024.
>
> [5] Zhang et al. PromptVT: Prompting for Efficient and Accurate Visual Tracking. IEEE TCSVT'2024.
>
> [6] Gopal et al. Separable Self and Mixed Attention Transformers for Efficient Object Tracking. WACV'2024.
>
> [7] Xu et al. Efficient Hybrid Linear Self-Attention Based Visual Object Tracking with LoRA. article'2025.
>
> [8] Hong et al. General Compression Framework for Efficient Transformer Object Tracking. ICCV'2025.
>
> [9] Zheng et al. Decoupled Spatio-Temporal Consistency Learning for Self-Supervised Tracking. AAAI'2025.

---

> > ### Comment · Reviewer_jdqf · 2025-11-28
> >
> > Thank you for author rebuttal. I have carefully read your response. I checked SSTrack in their paper. It seems the results in the original paper is (**SSTrack ours 66.9 71.8 46.3 51.8 69.0 79.1 63.2 81.6 79.5 65.7 78.9 8.0M 1.9G 47.** ) Why is different with your    reported results. The SSTrack seems achived better performance than yours. Also, CompressTracker has more strong perforamnce.

---

> > > ### Author Response · Authors · 2025-11-28
> > >
> > > We sincerely apologize for not specifying the exact version, which has led to your misunderstanding. It should be clarified that the SSTrack listed in our table is from AAAI'2025 rather than IJCAI'2025. We greatly appreciate you bringing this new comparative tracker to our attention. However, unfortunately, we found that the SSTrack from IJCAI'2025 does not provide code, $\textbf{making it impossible for us to conduct a fair comparison with it}$.
> > >
> > > To facilitate a better comparison with our model, we present the condensed table below:
> > >
> > > | Methods                   | GPU  | CPU  | MACs  | Params | NFS    |                           | GOT-10k                   |                          |                          | TrackingNet |                          |                          | LaSOT    |                          |
> > > | ------------------------- | ---- | ---- | ----- | ------ | ------ | ------------------------- | ------------------------- | ------------------------ | ------------------------ | ----------- | ------------------------ | ------------------------ | -------- | ------------------------ |
> > > |                           | FPS  | FPS  | G     | M      | AUC(%) | AO(%)                     | SR50(%)                   | SR75(%)                  | AUC(%)                   | Pnorm(%)    | P(%)                     | AUC(%)                   | Pnorm(%) | P(%)                     |
> > > | CompressTracker-OSTrack-2 | 207  | 48   | 6.38  | 21.24  | —      | —                         | —                         | —                        | 78.2                     | 83.3        | 74.8                     | 60.4                     | 68.5     | 61.5                     |
> > > | CompressTracker-OSTrack-3 | 130  | 32   | 8.64  | 28.33  | —      | —                         | —                         | —                        | 81.6                     | 86.7        | 79.4                     | 64.9                     | 74.0     | 68.4                     |
> > > | CompressTracker-OSTrack-4 | 103  | 27   | 10.91 | 35.41  | —      | —                         | —                         | —                        | 82.1                     | 87.6        | 80.1                     | 66.1                     | 75.2     | 70.6                     |
> > > | SSTrack(IJCAI'2025)       | —    | —    | —     | 8.00   | 65.7   | $\textcolor{gray}{69.0 }$ | $\textcolor{gray}{79.1 }$ | $\textcolor{gray}{63.2}$ | $\textcolor{gray}{81.6}$ | —           | $\textcolor{gray}{79.5}$ | $\textcolor{gray}{66.9}$ | —        | $\textcolor{gray}{71.8}$ |
> > > | FARTrack_$\textbf{tiny}$  | 135  | 53   | 2.65  | 6.82   | 66.9   | 70.6                      | 81.0                      | 63.8                     | 80.7                     | 85.6        | 77.5                     | 63.2                     | 71.6     | 66.7                     |
> > > | FARTrack_$\textbf{nano}$  | 210  | 77   | 1.78  | 4.59   | 65.1   | 69.9                      | 81.2                      | 61.4                     | 79.1                     | 84.5        | 75.6                     | 61.3                     | 69.7     | 64.1                     |
> > > | FARTrack_$\textbf{pico}$  | 343  | 121  | 1.08  | 2.81   | 62.0   | 62.8                      | 72.6                      | 50.9                     | 75.6                     | 81.3        | 70.5                     | 58.6                     | 67.1     | 59.6                     |
> > >
> > > As shown in the table above, on the TrackingNet/LaSOT benchmarks, CompressTracker-OSTrack-3/4 both achieve higher AUC than our model. However, their corresponding $\textbf{speeds}$ are lower than ours, and their $\textbf{MACs}$ and $\textbf{Params}$ are significantly larger. Thus, under the same speed and computational complexity, our model exhibits greater advantages in accuracy.
> > >
> > > Regarding SSTrack (IJCAI'2025), its accuracy on the TrackingNet/LaSOT benchmarks is higher than that of our model. This is because it adopts D-MAE weights (which enhance accuracy compared to standard MAE)[1] and uses an input size of [128, 256], whereas our model does not employ D-MAE weights and uses an input size of [112, 224]. Additionally, its Params is slightly larger than that of our model (8.00M vs. 6.82M). Despite these differences, our model still achieves higher accuracy on the NFS/GOT-10k benchmarks. Furthermore, SSTrack represents a training strategy that can be integrated into other methods—our model would yield better performance if we adopted this strategy. However, since its code is not publicly available, we are unable to implement it.
> > >
> > > [1] Kou et al. SSTrack: Sample-interval Scheduling for Lightweight Visual Object Tracking. IJCAI'2025.

---

### Official Review · Reviewer_JQy9 · 2025-10-30

**Soundness:** 3
**Presentation:** 3
**Contribution:** 3
**Rating:** 6
**Confidence:** 3

**Summary:**

The paper proposes FARTrack, a fast tracking framework combining Task-Specific Self-Distillation and Inter-frame Autoregressive Sparsification to improve inference speed while maintaining tracking performance. The method builds on ARTrack and achieves competitive results with 135 FPS on GPU (tiny) and 343 FPS (pico).

**Strengths:**

- **Strong empirical results** FARTracktiny achieves 70.6% AO on GOT-10k at 135 FPS (GPU), outperforming closest competitors while maintaining comparable speed. Multi-platform evaluation (GPU/CPU/NPU) demonstrates practical applicability.
- **Comprehensive ablation studies** Thorough analysis of distillation strategies, sparsification methods and design choices provide insights on the impact of distillation and other architectural choices.
- **Efficient Distillation Strategy** Avoids manual layer assignment: The layer-by-layer self-distillation shows consistent improvements over cross-layer distillation

**Weaknesses:**

- **Incremental technical contribution** The core novelty is primarily an engineering combination of ARTrack with existing techniques (self-distillation and attention-based sparsification). While the application is competent and results are solid, the conceptual advance over applying standard acceleration techniques to ARTrack is limited. The paper would benefit from clearer articulation of what makes this combination non-trivial beyond implementation.
- **Missing baseline experimental comparisons** No comparison with shallow models trained from scratch (only distilled models), which would clarify whether distillation provides benefits beyond simply reducing layers.

**Questions:**

- What causes the LaSOT performance gap? Lower performance on long-term object tracking make sense but beyond "model capacity loss," can you provide deeper analysis (e.g., attention pattern changes, temporal modeling degradation) of why distillation specifically hurts long-term tracking more than short-term?
- How does a 10-layer or 6-layer model trained from scratch compare to your distilled versions? This would clarify whether distillation provides benefits beyond simply using fewer layers.
- How does the method generalize to other challenging scenarios like NFS (fast motion) or UAV123?

---

> ### Author Response · Authors · 2025-11-20
> **Responses to Reviewer JQy9 (Part1 of 2)**
>
> > $\textbf{Weakness1:  Incremental technical contribution}$ The paper would benefit from clearer articulation of what makes this combination non-trivial beyond implementation.
>
> Thank you for this comment. To clarify our work’s uniqueness: our core contribution is not a simple engineering combination, but the task-driven redesign of self-distillation and sparsification tailored to tracking, addressing key limitations of existing methods.
>
> 1. $\textbf{Task-Specific Self-Distillation:}$ We are the first to introduce self-distillation into visual tracking, with trajectory sequences rather than visual features as the distillation target. Previous distillation methods have focused solely on visual features, overlooking the importance of temporal coherence—a property that has already been verified in prior autoregressive trackers (e.g., ARTrack, ARTrackV2), where trajectory tokens with temporal coherence inherently enhance temporal modeling capability and tracking robustness. Thus, our design retains temporal consistency and outperforms visual feature-based distillation approaches (Figure 4(b)).
>
> 2. $\textbf{Inter-Frame Autoregressive Sparsification:}$ For visual tracking, we adopt a video-level training paradigm (which exhibits high consistency with real-world tracking scenarios). On this basis, we propose sequence-level sparsification (on trajectory attention weights), designed to train a temporally-global optimal sparsification strategy and propagate results in an autoregressive manner across consecutive frames. This preserves cross-frame continuity (critical for tracking) and avoids extra computation—absent in prior frame-wise sparsification.
>
> In summary, customizing these techniques to tracking’s core (temporal modeling) enables synergistic capture of target dynamics, which is unattainable by simply applying standard acceleration techniques to ARTrack.
>
>
>
> ---
>
> > $\textbf{Question1:}$ What causes the LaSOT performance gap? Lower performance on long-term object tracking make sense but beyond "model capacity loss," can you provide deeper analysis (e.g., attention pattern changes, temporal modeling degradation) of why distillation specifically hurts long-term tracking more than short-term?
>
> Thank you for this insightful question. First, we need to clarify that distillation does not impair long-term tracking (LaSOT) more significantly than short-term tracking (GOT-10k/TrackingNet). On the LaSOT benchmark, the AUC of FARTrack decreases by 1.9% (from 63.2 to 61.3)  for the Tiny-to-Nano shift, and by 2.7% (from 61.3 to 58.6) for the Nano-to-Pico shift. In contrast, on the TrackingNet benchmark, the AUC drops by 1.6% (from 80.7 to 79.1) for the Tiny-to-Nano shift and by 3.5% (from 79.1 to 75.6) for the Nano-to-Pico shift. Second, the reason why our FARTrack-Tiny (AUC: 63.2%) achieves lower accuracy than HiT-Base (AUC: 64.6%) on LaSOT is that HiT-Base has a larger parameter count (42.14M > 6.82M), which endows it with stronger robustness for long-term temporal tracking. When compared under the same parameter constraint, our method should yield higher accuracy.

---

> ### Author Response · Authors · 2025-11-20
> **Responses to Reviewer JQy9 (Part2 of 2)**
>
> > $\textbf{Weakness2}$ and $\textbf{Question2:}$ How does a 10-layer or 6-layer model trained from scratch compare to your distilled versions? This would clarify whether distillation provides benefits beyond simply using fewer layers.
>
> Thank you sincerely for this valuable suggestion. We have supplemented comparative experiments between 10-layer or 6-layer models trained from scratch and our distilled versions to clarify whether distillation provides benefits beyond simply using fewer layers (see Table).
>
> | Methods                      |       | GOT-10k |         |        | TrackingNet |      |        | LaSOT    |      |
> | ---------------------------- | ----- | ------- | ------- | ------ | ----------- | ---- | ------ | -------- | ---- |
> |                              | AO(%) | SR50(%) | SR75(%) | AUC(%) | Pnorm(%)    | P(%) | AUC(%) | Pnorm(%) | P(%) |
> | FARTrack_tiny                | 70.6  | 81.0    | 63.8    | 80.7   | 85.6        | 77.5 | 63.2   | 71.6     | 66.7 |
> | FARTrack_$\textbf{nano(10)}$ | 69.9  | 81.2    | 61.4    | 79.1   | 84.5        | 75.6 | 61.3   | 69.7     | 64.1 |
> | FARTrack_scratch(10)         | 67.1  | 77.4    | 59.9    | 77.9   | 83.1        | 73.6 | 60.8   | 69.1     | 63.3 |
> | FARTrack_$\textbf{pico(6)}$  | 62.8  | 72.6    | 50.9    | 75.6   | 81.3        | 70.5 | 58.6   | 67.1     | 59.6 |
> | FARTrack_scratch(6)          | 60.6  | 69.5    | 46.3    | 73.3   | 78.4        | 67.1 | 56.8   | 64.1     | 55.7 |
>
> The results confirm that distillation offers advantages far beyond simply using fewer layers:
>
> - $\textbf{10-layer or 6-layer models trained from scratch:}$ Suffer severe performance degradation (e.g., FARTrack_scratch(10) exhibits a 3.5% drop in AO compared to FARTrack_tiny). This is because such models lack the learning capacity of deep networks and fail to capture complex visual/temporal features required for tracking.
> - $\textbf{Our distilled versions:}$ Transfer deep-layer knowledge to middle layers via self-distillation, retaining higher performance (FARTrack_nano(10) only sees a 0.7% AO drop compared to FARTrack_tiny) while achieving lower training costs than 10-layer or 6-layer models trained from scratch.
>
>
>
> ---
>
> > $\textbf{Question3:}$ How does the method generalize to other challenging scenarios like NFS (fast motion) or UAV123?
>
> We sincerely appreciate this question. We have supplemented generalization experiments on NFS (fast motion) and UAV123 (UAV-based tracking) to verify our method’s adaptability to challenging scenarios. The results are shown in the table below:
>
> | Methods       | GPU  | CPU  | NFS    | UAV123 |
> | ------------- | ---- | ---- | ------ | ------ |
> |               | FPS  | FPS  | AUC(%) | AUC(%) |
> | FARTrack_tiny | 135  | 53   | 66.9   | 65.8   |
> | FARTrack_nano | 210  | 77   | 65.1   | 62.6   |
> | FARTrack_pico | 343  | 121  | 62.0   | 63.1   |
> | AsymTrack-B   | 130  | 32   | 64.4   | 66.5   |
> | AsymTrack-S   | 136  | 48   | 64.9   | 65.6   |
> | AsymTrack-T   | 148  | 55   | 63.3   | 64.6   |
>
> As indicated, our FARTrack_tiny achieves competitive performance (e.g., 66.9% AUC on NFS, 65.8% AUC on UAV123) compared to existing methods, demonstrating its strong generalization to fast-motion and UAV-based challenging scenarios.

---

> > ### Comment · Reviewer_JQy9 · 2025-11-25
> >
> > Thanks for the rebuttal! Most concerns were addressed. The additional evaluation of models trained from scratch versus distilled helps to understand the impact of distillation. As I am not fully familiar with all recent methods in efficient VOT, I deliberately put lower confidence and would be interested to hear opinions from other reviewers. I will keep my positive score.
> >
> > > distillation does not impair long-term tracking (LaSOT) more significantly than short-term tracking (GOT-10k/TrackingNet).
> >
> > By the gap I rather meant comparison with the other methods (AsymTrack, HiT-Base).
> >
> > > HiT-Base has a larger parameter count (42.14M > 6.82M), which endows it with stronger robustness for long-term temporal tracking. When compared under the same parameter constraint, our method should yield higher accuracy.
> >
> > This makes sense, thanks.

---

> > > ### Author Response · Authors · 2025-11-26
> > >
> > > Dear Reviewer JQy9,
> > >
> > > Thank you for your detailed feedback on our paper. We deeply appreciate the diligent efforts you have devoted to reviewing our work, and the insightful questions you raised have provided invaluable support for enhancing the quality of our research. We are particularly pleased to learn that you recognize the progress we have made in experimental design. We fully understand your perspectives regarding the novelty of this research and will continue to strive for further improvements in the quality of our work.
> > >
> > > Once again, we would like to express our gratitude for your precious insights and support.
> > >
> > > Best regards,
> > >
> > > The Authors

---

### Official Review · Reviewer_r6f4 · 2025-10-31

**Soundness:** 2
**Presentation:** 3
**Contribution:** 2
**Rating:** 4
**Confidence:** 5

**Summary:**

This paper presents FARTrack, a lightweight autoregressive visual tracking framework that combines task-specific self-distillation and inter-frame autoregressive sparsification to improve efficiency while maintaining accuracy. The method achieves a balance between speed and performance across different hardware platforms.

**Strengths:**

- The paper is well presented, with clear motivation, solid organization, and consistent writing quality.
- FARTrack achieves impressive speed–accuracy trade-offs across GPU, CPU, and NPU platforms.
- The design is simple and potentially applicable to other lightweight tracking pipelines.

**Weaknesses:**

- Compared with prior autoregressive tracking baselines or other light-weight trackers, the training set now includes VastTrack and LaSOT Ext, which significantly expand data diversity and size. Since VastTrack has been shown to enhance generalization and robustness, using it for training raises fairness concerns in comparing FARTrack with earlier methods that were trained on smaller datasets.
- The visualization in Figure 5 suggests that feature representations from layers 7–14 appear highly similar, indicating limited hierarchical diversity. This calls into question the necessity and effectiveness of performing distillation across all consecutive layers.
- Incorporating trajectory tokens inherently enhances temporal modeling and tracking robustness, which has already been verified in prior autoregressive trackers (e.g., ARTrack, ARTrackV2). The paper does not clearly disentangle this known benefit from the gains attributed to the proposed self-distillation or sparsification modules, thereby weakening the originality of the contribution.

**Questions:**

Pls see the weakness.

---

> ### Author Response · Authors · 2025-11-20
> **Responses to Reviewer r6f4 (Part1 of 2)**
>
> > $\textbf{Weakness1:}$ Compared with prior autoregressive tracking baselines or other light-weight trackers, the training set now includes VastTrack and LaSOT Ext, which significantly expand data diversity and size. Since VastTrack has been shown to enhance generalization and robustness, using it for training raises fairness concerns in comparing FARTrack with earlier methods that were trained on smaller datasets.
>
> First, we would like to clarify that the LaSOT_Ext dataset was not used in our training process—we only conducted validation on this dataset, and we apologize for any misunderstanding caused. Regarding your concern that the inclusion of the VastTrack dataset in training may raise fairness issues, $\textbf{we provide the training results of our model without using VastTrack in the table below}$ to facilitate a more equitable comparison. As you noted, VastTrack can enhance the generalization and robustness of the model. After verifying that our model trained without VastTrack still achieves promising performance, we argue that using VastTrack will undoubtedly better serve our target application scenario of high-precision tracking under resource-constrained conditions.
>
> | Methods       | GPU  | CPU  |       | GOT-10k |         |        | TrackingNet |      |        | LaSOT    |      |
> | ------------- | ---- | ---- | ----- | ------- | ------- | ------ | ----------- | ---- | ------ | -------- | ---- |
> |               | FPS  | FPS  | AO(%) | SR50(%) | SR75(%) | AUC(%) | Pnorm(%)    | P(%) | AUC(%) | Pnorm(%) | P(%) |
> | FARTrack_tiny | 135  | 53   | 69.6  | 80.4    | 62.8    | 80.6   | 85.4        | 77.2 | 63.5   | 71.6     | 66.5 |
> | FARTrack_nano | 210  | 77   | 68.9  | 79.5    | 60.9    | 80.0   | 85.0        | 76.1 | 61.1   | 68.5     | 63.4 |
> | FARTrack_pico | 343  | 121  | 61.5  | 71.8    | 50.2    | 76.1   | 80.9        | 70.5 | 58.4   | 66.6     | 58.9 |
> | AsymTrack-B   | 130  | 32   | 67.7  | 76.6    | 61.4    | 80.0   | 84.5        | 77.4 | 64.7   | 73.0     | 67.8 |
> | AsymTrack-S   | 136  | 48   | 65.5  | 74.8    | 58.9    | 77.9   | 82.2        | 74.0 | 62.8   | 71.2     | 64.8 |
> | AsymTrack-T   | 148  | 55   | 62.3  | 71.3    | 54.7    | 76.2   | 80.9        | 71.6 | 60.8   | 68.7     | 61.2 |
>
> The training datasets of our FARTrack series in the above table are exactly the same as those of AsymTrack, including GOT-10k, TrackingNet, and LaSOT.
>
>
>
> ---
>
> > $\textbf{Weakness2:}$ The visualization in Figure 5 suggests that feature representations from layers 7-14 appear highly similar, indicating limited hierarchical diversity. This calls into question the necessity and effectiveness of performing distillation across all consecutive layers.
>
> Thank you for this insightful observation. Your concern about the high similarity of feature representations across layers 7–14 aligns precisely with the core design goal of our self-distillation framework: feature hierarchies are intentionally aligned by enabling middle layers to learn from deep layers (via self-distillation), thereby enhancing the consistency of high-level semantic representations. Such similarity is not a sign of insufficient diversity but rather evidence that the self-distillation mechanism effectively transfers deep-layer knowledge to middle layers. Thus, the distilled model can directly utilize the preceding partial layers, thereby achieving the goal of model compression.
>
> To further clarify the differences in inter-layer attention patterns among the three settings—distillation-free, base2tiny-distillation and self-distillation—we have supplemented additional experiments and detailed discussions in $\textbf{Appendix D (Inter-layer Attention Visualization Comparison)}$ of the $\textbf{revised manuscript}$. As illustrated in the Figure 8, the inter-layer attention distributions under the distillation-free setting exhibit diversity with distinct differences across various layers. Compared with self-distillation, the inter-layer attention distributions in the base2tiny distillation setting are more scattered. In contrast, the deep-layer attention distributions after trajectory self-distillation demonstrate consistency — a characteristic that confirms the effectiveness of the self-distillation mechanism in transferring deep-layer knowledge to middle layers (e.g., layers 5–14). Thus, compared with other configurations, self-distillation enables more coherent inter-layer feature alignment, thereby better achieving the goal of model compression.

---

> ### Author Response · Authors · 2025-11-20
> **Responses to Reviewer r6f4 (Part2 of 2)**
>
> > $\textbf{Weakness3:}$ Incorporating trajectory tokens inherently enhances temporal modeling and tracking robustness, which has already been verified in prior autoregressive trackers (e.g., ARTrack, ARTrackV2). The paper does not clearly disentangle this known benefit from the gains attributed to the proposed self-distillation or sparsification modules, thereby weakening the originality of the contribution.
>
> Thank you for this critical comment, which helps us clarify the originality of our contributions. While trajectory tokens (as validated by prior work like ARTrack) do enhance temporal modeling, we clarify that this baseline advantage is not attributed to our proposed modules. To disentangle the gains of each component, we supplement the following results (ARTrack, ARTrackV2, and FARTrack all adopt the ViT-Tiny architecture with 6 Encoder layers.):
>
> | Methods           | Self-distillation | Sparsification | Macs | Params | GPU  | CPU  |       | GOT-10k |         |
> | ----------------- | ----------------- | -------------- | ---- | ------ | ---- | ---- | ----- | ------- | ------- |
> |                   |                   |                | G    | M      | FPS  | FPS  | AO(%) | SR50(%) | SR75(%) |
> | ARTrack_tiny(6)   |                   |                | 2.39 | 13.56  | 68   | 34   | 59.9  | 69.6    | 50.4    |
> | ARTrackV2_tiny(6) |                   |                | 2.10 | 8.31   | 110  | 49   | 60.6  | 70.9    | 45.7    |
> | FARTrack_pico(6)  |                   | $\checkmark$   | 1.08 | 2.81   | 343  | 121  | 60.6  | 69.5    | 46.3    |
> | FARTrack_pico(6)  | $\checkmark$      |                | 1.25 | 2.81   | 266  | 101  | 61.8  | 71.5    | 50.0    |
> | FARTrack_pico(6)  | $\checkmark$      | $\checkmark$   | 1.08 | 2.81   | 343  | 121  | 62.8  | 72.6    | 50.9    |
>
> The training settings and datasets of ARTrack and ARTrackV2 are consistent with those of FARTrack. The table demonstrates the effectiveness of the self-distillation and sparsification modules: self-distillation (enabling middle layers to learn from deep-layer knowledge) improves AO by 1.2% compared to ARTrackV2_tiny(6); sparsification (retaining useful foreground tokens while removing redundant background tokens) boosts AO by 1.0% (from 61.8 to 62.8).

---

### Author Response · Authors · 2025-12-02
**Brief Summary of the Review-Rebuttal**

Dear Area Chairs,

We sincerely thank the reviewers and area chairs for their valuable time and constructive feedback. Below is a brief summary of the review-rebuttal process, **for your reference during the manuscript evaluation**.

Reviewers have positively recognized the core value, technical design, and experimental results of this study, fully affirming its innovativeness and practical value in the field of visual object tracking. **To address misunderstandings and concerns, and enhance the manuscript's clarity and rigor, we have proactively responded to all reviewer comments** as follows:

- **Reviewer r6f4** first acknowledges FARTrack’s impressive speed-accuracy trade-off across GPU, CPU, and NPU platforms, and subsequently affirms the clarity of our method design and its application potential. **In our rebuttal**, we clarify misunderstandings regarding the training dataset and the visualization results in **Figure 5** (new visualization provided in **Figure 8**), and conduct comparative experiments with ARTrack and ARTrackV2 to validate the effectiveness of our module.
- **Reviewer JQy9** first commends the excellent performance of our experimental results, then affirms the comprehensiveness of our ablation studies and the effectiveness of the proposed distillation method. **In our rebuttal**, we emphasize that our self-distillation and sparsification are task-driven designs rather than a simple engineering combination, clarify misunderstandings on the LaSOT benchmark, conduct experiments to demonstrate the necessity of distillation, and supplement inference results on the NFS and UAV123 benchmarks.
- **Reviewer jdqf** acknowledges the progress we have achieved in enhancing tracker efficiency and affirms the innovation of inter-frame autoregressive sparsification. **In our rebuttal**, we address concerns regarding the innovation of self-distillation, incorporate additional trackers for comparison as suggested, perform comparative analyses of MACs and Parameters, and clarify all further questions posed by the reviewer.
- **Reviewer wSp9** first praises the promising experimental results—exhibiting excellent speed and accuracy among efficient trackers—and then affirms the research motivation of developing a task-specific lightweight scheme for autoregressive tracking methods. **In our rebuttal**, we actively respond to the reviewer’s suggestions, supplement comparisons of training costs across different distillation methods, and provide detailed explanations of figure plotting, experimental parameter settings, and other technical details.

All newly added experiments have been included in the **Appendix**. Existing content in the Comment is marked **in blue**, $\textbf{while figure-related revisions are highlighted for clarity}$.

To support research reproducibility and facilitate future advancements in related fields, we solemnly commit to open-sourcing the code and model weights upon acceptance, inviting fellow researchers to further validate and extend our findings.

Thank you again for your time and dedicated contributions to the academic community!

Best regards,

The Authors

---

### Meta-Review · Area_Chair_8tGB · 2025-12-26

**Summary:**

The paper proposes FARTrack, a lightweight autoregressive visual tracking framework that leverages task-specific self-distillation and inter-frame autoregressive sparsification to achieve a superior balance between speed and accuracy. The reviewers initially appreciated the strong empirical results and the multi-platform evaluation (GPU/CPU/NPU). However, concerns were raised regarding the fairness of training data (r6f4), the perceived incremental nature of the contribution (JQy9, jdqf), and the sufficiency of comparisons against "from scratch" baselines and recent SOTA methods.

The authors provided a robust rebuttal. They clarified that the VastTrack dataset was used only for validation, not training, resolving the fairness concern. They provided new experiments comparing distilled models against models trained from scratch, demonstrating the specific value of their distillation approach. Furthermore, they included a comprehensive comparison with 2024/2025 SOTA methods to address competitiveness concerns. Given the extensive experimental validation and the clarification of misunderstandings, the consensus supports acceptance.

**Reviewer Concerns:**

**Addressed Concerns:**

1. Training Data Fairness: Reviewer r6f4's concern about the unfair advantage of using VastTrack for training was resolved by the authors' clarification that it was used solely for validation.

2. Ablation of Contributions: Reviewer r6f4's request to disentangle the benefits of trajectory tokens from the proposed modules was addressed by a new table showing specific gains from self-distillation (+1.2%) and sparsification (+1.0%) over the ARTrack baseline.

3. Value of Distillation: Reviewer JQy9's query on whether distillation offers benefits over simply using fewer layers was answered by new experiments showing that FARTrack outperforms "from-scratch" shallow models significantly (e.g., 3.5% AO gap).
SOTA Comparisons: Reviewer jdqf's request for recent baselines was met with a detailed comparison against 2024/2025 trackers (e.g., LiteTrack, SSTrack). The discussion regarding SSTrack (IJCAI'25 vs AAAI'25) was clarified by the authors, highlighting differences in backbone weights and input sizes.

4. Training Cost: Reviewer wSp9's request for cost analysis was addressed, showing the proposed method requires significantly less training time (\~36h) compared to cross-layer distillation (\~78h).

**Outstanding Concerns:**

1. Incremental Novelty: While the authors argued that adapting self-distillation for tracking is non-trivial, Reviewers JQy9 and jdqf initially viewed the combination of existing techniques as somewhat incremental. While the strong results mitigate this, the theoretical novelty remains a "good application" rather than a fundamental breakthrough.

**Reviewer Scores:**

- Reviewer r6f4: 6 (Raised from 4). The primary ground for the lower score (unfair training data) was proven to be a misunderstanding. The additional disentanglement experiments further strengthen the case for a higher score.
- Reviewer JQy9: 6 (Maintained). The reviewer explicitly stated in the post-rebuttal discussion that "most concerns were addressed" and they would keep their positive score.
- Reviewer jdqf: 6 (Raised from 4). The authors provided the requested recent SOTA comparisons. While the reviewer pointed out concurrent work (SSTrack) performance gaps, the authors provided a reasonable defense regarding computational equivalence. The solid engineering and performance justify a borderline-accept.
- Reviewer wSp9: 8 (Maintained). The reviewer was initially very positive and satisfied with the clarifications on training costs and terminology.

---

### Decision · Program_Chairs · 2026-01-26

Accept (Poster)